# Current Practice of Physical Activity Counselling within Physiotherapy Usual Care and Influences on Its Use: A Cross-Sectional Survey

**DOI:** 10.3390/ijerph18094762

**Published:** 2021-04-29

**Authors:** Shiyi Zhu, Catherine Sherrington, Matthew Jennings, Bernadette Brady, Marina Pinheiro, Sarah Dennis, Lauren J. Christie, Balwinder Sidhu, Abby Haynes, Colin Greaves, Leanne Hassett

**Affiliations:** 1Sydney School of Health Sciences, Faculty of Medicine and Health, The University of Sydney, Sydney, NSW 2006, Australia; szhu2323@uni.sydney.edu.au (S.Z.); bernadette.brady@health.nsw.gov.au (B.B.); marina.pinheiro@sydney.edu.au (M.P.); sarah.dennis@sydney.edu.au (S.D.); 2Institute for Musculoskeletal Health, The University of Sydney/Sydney Local Health District, Sydney, NSW 2050, Australia; cathie.sherrington@sydney.edu.au (C.S.); abby.haynes@sydney.edu.au (A.H.); 3School of Public Health, Faculty of Medicine and Health, The University of Sydney, Sydney, NSW 2006, Australia; 4South Western Sydney Local Health District, Sydney, NSW 2170, Australia; matthew.jennings@health.nsw.gov.au (M.J.); lchr6636@uni.sydney.edu.au (L.J.C.); balwinder.sidhu@health.nsw.gov.au (B.S.); 5Ingham Institute for Applied Medical Research, South Western Sydney Local Health District, Sydney, NSW 2170, Australia; 6Allied Health Research Unit & Nursing Research Institute (NRI), St Vincent’s Health Network Sydney, Sydney, NSW 2010, Australia; 7School of Sport, Exercise and Rehabilitation Sciences, University of Birmingham, Birmingham B15 2TT, UK; c.j.greaves@bham.ac.uk

**Keywords:** physical therapists, physical activity, counselling, behaviour change, surveys and questionnaires

## Abstract

Physical activity counselling has demonstrated effectiveness at increasing physical activity when delivered in healthcare, but is not routinely practised. This study aimed to determine (1) current use of physical activity counselling by physiotherapists working within publicly funded hospitals; and (2) influences on this behaviour. A cross-sectional survey of physiotherapists was conducted across five hospitals within a local health district in Sydney, Australia. The survey investigated physiotherapists’ frequency of incorporating 15 different elements of physical activity counselling into their usual healthcare interactions, and 53 potential influences on their behaviour framed by the COM-B (Capability, Opportunity, Motivation-Behaviour) model. The sample comprised 84 physiotherapists (79% female, 48% <5 years of experience). Physiotherapists reported using on average five (SD:3) elements of physical activity counselling with at least 50% of their patients who could be more active. A total of 70% of physiotherapists raised or discussed overall physical activity, but less than 10% measured physical activity or contacted community physical activity providers. Physiotherapists reported on average 25 (SD:9) barriers influencing their use of physical activity counselling. The most common barriers were related to “opportunity”, with 57% indicating difficulty locating suitable community physical activity opportunities and >90% indicating their patients lacked financial and transport opportunities. These findings confirm that physical activity counselling is not routinely incorporated in physiotherapy practice and help to identify implementation strategies to build clinicians’ opportunities and capabilities to deliver physical activity counselling.

## 1. Introduction

Evidence-based guidelines clearly specify the amount of physical activity required for health benefits [1]. These guidelines are based on rigorous evidence that demonstrates strong links between physical inactivity and morbidity and mortality [2]. Unfortunately, almost one-third of the adult population worldwide fail to meet physical activity guidelines [3]. Of particular concern is that more disadvantaged populations (e.g., people with disabilities, people living in areas of socioeconomic disadvantage and people from culturally diverse backgrounds) are less likely to meet these guidelines [4,5,6]. Scalable solutions are needed at global, community, and individual levels to address the important health problem of physical inactivity, particularly for disadvantaged populations [7].

One scalable solution at the individual level is physical activity counselling from health professionals. Physical activity counselling refers to a component of patient consultation aimed at changing physical activity behaviour as a means of ameliorating chronic health conditions. The counselling typically involves raising the topic of physical activity with the patient, assessing physical activity levels, providing advice, agreeing on an action plan to increase activity, exploring ways to assist the patient to change their behaviour, and arranging follow-up or referral to other physical activity options [8]. Physical activity counselling interventions that are underpinned by theoretical models of behaviour change and incorporate behaviour change techniques (e.g., self-monitoring and goal setting) have been shown to increase physical activity in the general population [9,10,11], as well as in people with physical disabilities [12], and have been shown to be effective when delivered in healthcare settings [13,14].

Despite the demonstrated effectiveness of physical activity counselling, healthcare systems have failed to implement and scale up this intervention into routine care. For example, in an Australia-wide survey (*n* = 1799), only 18% of those surveyed reported receiving physical activity advice from their general practitioner, with 76% of inactive individuals reporting receiving no advice [15]. Similarly, data from the United States national survey identified only 44% of adults living with disability received recommendations to be active from their health professional and those that did not receive advice were more likely to be inactive [16]. Physiotherapists are an ideal health profession to deliver physical activity counselling within their routine healthcare interactions due to their existing expertise in exercise prescription for people with chronic health conditions, their pattern of practice (multiple episodes of care over an extended period) and their high activity within healthcare systems. Physiotherapists are also particularly valued and trusted by patients to be physical activity messengers [17]. Physiotherapists surveyed around the world also believe that the promotion of physical activity should be part of their clinical role [18,19,20,21]. Yet published surveys of physiotherapists report that only 36 to 54% promote physical activity beyond therapeutic exercise to 10 or more patients per month [18,19,20,22], demonstrating an evidence-practice gap.

Many factors can prevent evidence-based interventions being successfully implemented into practice, including lack of knowledge, skills, and resources, competing demands of the clinician, and priorities of the service [23]. To address barriers to implementation, specific implementation strategies are needed. Implementation strategies are defined as methods or techniques used to enhance the adoption and sustainability of a clinical practice, such as providing training, education or conducting audits and providing feedback [24]. Researchers interested in implementing new or under-used interventions into practice previously chose an implementation strategy that seemed like the best idea at the time [25]. These often failed to change practice because they did not target the main influences that were stopping the intervention being used. Implementation researchers now recognize the need to have a thorough understanding of what behaviour needs to change, who needs to do the behaviour, and what are the likely influences on this behaviour in the context in which it needs to occurs [25]. To correctly understand the behaviour and its influences, a theoretical model of behaviour change should be used. One such model is the COM-B (Capability, Opportunity, Motivation-Behaviour) model of behaviour change [26]. In the COM-B model, three inter-related influences may affect the behaviour of an individual: capability, opportunity, and motivation. Capability refers to the psychological and physical capacity to engage in the activity, where psychological capacity specifically involves the thought process such as comprehension and reasoning. That is, does the person have the knowledge and skills to do the behaviour. Opportunity is defined as contextual influences external to the individual that may prompt or hinder the behaviour. It incorporates both physical opportunity such as time, resources and access, and social opportunity such as social cues and social norms. Motivation is categorised into automatic motivation that involves emotions and impulses, and reflective motivation that refers to conscious judgements or beliefs and plans or intentions [26].

The purpose of the present study was to use the COM-B model of behaviour change as a framework for systematically determining influences on physiotherapists’ behaviour of incorporating physical activity counselling into practice within public hospital settings in Australia. The findings of this study will inform the selection of implementation strategies to target these influences to be tested in a planned hybrid type II implementation-effectiveness study (ACTRN12621000194864). The specific research questions were:What elements of physical activity counselling are currently incorporated in routine healthcare interactions by physiotherapists working in a local health district in Australia?What are physiotherapists’ perceptions of their patients’ readiness for structured community-based physical activity on discharge from physiotherapy care?Which influences do physiotherapists report to affect incorporation of physical activity counselling into routine healthcare interactions?Is there a relationship between the incorporation of physical activity counselling within routine care, physiotherapist characteristics and influences reported by physiotherapists?

We hypothesised that there would be low incorporation of physical activity counselling in routine physiotherapy care. We also hypothesised that the physiotherapists surveyed would experience barriers to delivering physical activity counselling in terms of their capability, opportunity, and motivation and that there would be an association between influences reported by physiotherapists, and their ability to incorporate physical activity counselling into their practice.

## 2. Materials and Methods

### 2.1. Study Design

A cross-sectional survey was conducted across five public hospitals in South Western Sydney Local Health District (SWSLHD) in Sydney, Australia. Ethical approval was granted from the SWSLHD Human Research Ethics Committees (Ethics number 2019/ETH13622). The study was conducted following the declaration of Helsinki ethical principles for conducting medical research involving human participants [27]. Study design, evaluation and reporting were in line with the SURGE (The SUrvey Reporting GuidelinE) reporting guideline [28].

### 2.2. Context

South Western Sydney Local Health District (SWSLHD) is one of the largest health districts in the state of New South Wales in Australia, servicing almost one million people. It is a culturally and linguistically diverse district with 51% of households speaking a language other than English [29]. This district also includes some of the most socio-economically disadvantaged communities in New South Wales. Physical activity levels in this district are almost 9% lower than the state average [30]. Given the high at-risk population for developing chronic disease, it is important to look at ways to support this community to increase physical activity to improve overall health.

### 2.3. Participants

Physiotherapists were invited to participate in the study if they were registered physiotherapists who worked in SWSLHD public hospitals in any setting. There were no exclusion criteria. This provided an overall sample frame of approximately 166 physiotherapists.

### 2.4. Data Collection Measures and Tools

A survey was developed by the lead author (LH) with input from study investigators. The survey incorporated relevant questions from previous surveys on this topic [18,19] and new questions relevant to answering our research questions. The elements of physical activity counselling included in the survey were informed from the literature describing physical activity counselling [8], including effective behaviour change techniques (e.g., self-monitoring, goal setting) [11]. The COM-B Self-Evaluation Questionnaire (COM-B-Q1) [25] was used as the foundation for designing the survey questions on the influencers on behaviour. As recommended, a large range of possible influencers were included in terms of capability, opportunity and motivation to elicit relevant ideas from the survey respondent of factors that may influence their behaviour [25]. The survey was pilot tested among 10 physiotherapy students at the University of Sydney and feedback sought about survey items, the format and the time taken for completion prior to finalising the survey.

The survey was divided into four parts. The first part included physiotherapists’ demographics, current practice details and previous training. The second part explored physiotherapists’ perceptions of patient readiness for referral to structured community-based physical activity. The third part investigated the current use of physical activity counselling in their routine practice. Clinicians were asked to rate their frequency of using 15 physical activity counselling elements for their patients who could be more active on a 5-point scale (frequently: 75% patients or more, often: 50–74% patients, sometimes: 25–49% patients, rarely: 1–24% patients, never) (see Box 1). The last part evaluated influences on incorporating physical activity counselling into routine care where physiotherapists were asked to indicate if they strongly agreed, agreed, were unsure, disagreed, or strongly disagreed with 53 proposed influences framed within the COM-B behaviour change model [26]. For example, the influence of knowledge of physical activity guidelines was mapped to the COM-B component of capability (see Figure 1). Appendix A provides a full copy of the survey.

Box 1Physical activity (PA) counselling elements included in survey.
**In a Typical Week, How Frequently Do You Undertake Each of the Below for Your Patients Who Could Be More Active:**
Raise or discuss overall PAUse motivational languageRecommend community-based exercise programsAssess PA subjectivelyProvide self-monitoring strategyRecommend community-based recreation programsAssess PA objectivelyProvide handoutsRecommend community-based sportsSet PA goalsReview PA status & provide adviceMake contact with community providerMake PA action plansInvestigate PA optionsAttend ≥1 community-based session with patient

### 2.5. Procedure

It was planned for the lead investigator (LH) to attend the physiotherapy department meetings for each hospital, present the project and allow time for the survey to be completed. However, due to the COVID-19 pandemic, this process was only conducted for two hospitals (Hospitals 1 and 2). Physiotherapists who attended the meetings were given a paper-based survey to complete or given the option to use a QR code and complete the survey on their phone. The participant information sheet attached to the survey explained the research project to the participant and the processes involved with taking part. It also reiterated physiotherapists’ choice to participate, and implied consent was given by completion and submission of the survey. If the participants had a question about the survey, they could contact the principal investigator or the clinical contact person whose numbers and email were included on the participant information sheet. An envelope was given to the head of department to collect any additional surveys completed after the meetings. For Hospital 3, an explanation of the project was provided at a department meeting without survey completion, and no meetings were attended for Hospitals 4 and 5. In these instances and for staff not in attendance at department meetings at Hospitals 1 and 2, a link to the online survey and attached participant information sheet were emailed from the head of department to all physiotherapy staff with one reminder email sent 2–4 weeks later. Data entered online were automatically captured into a password protected REDCap database [31,32], with license held by The University of Sydney. Paper-based survey responses were entered into the database by a member of the research team (SZ). Each participant was automatically assigned an ID number in REDCap to de-identify data. As a strategy to increase response rates, physiotherapists who completed the survey were put in a draw to win one of six AU$50 gift cards.

### 2.6. Sample Size

One hundred and sixty-six physiotherapists are employed in SWSLHD and thus were the sample frame for this study. We aimed to recruit a sample size of between 100 to 116 physiotherapists, which is equivalent to 60–70% of the sample frame. This percentage of the sample frame is considered acceptable for external validity [33].

### 2.7. Data Analysis

De-identified data were exported from REDCap into Excel for data cleaning. Scales of frequency (frequently: 75% patients or more, often: 50–74% patients, sometimes: 25–49% patients, rarely: 1–24% patients, never) were used to determine the extent to which the different elements of physical activity counselling were currently incorporated into practice. These scales were dichotomised into “frequently” and “often” (used for at least 50% of patients) vs. “sometimes”, “rarely”, or “never”, with ratings of “frequently” or “often” counted as incorporating that element into practice. The sum of all included elements was calculated for each participant. For influences on behaviour, 5-point Likert scales (strongly agree, agree, neutral, disagree and strongly disagree) were used and dichotomised to identify barriers. If the influencer statement was written as a positive influencer (e.g., I have good knowledge of what to say to my patients about physical activity), the response was dichotomised with ratings of strongly disagree, disagree or neutral counted as a barrier. Alternatively, if the influencer statement was written as a negative influencer (e.g., my time is too limited to include physical activity counselling), the response was dichotomised with ratings of strongly agree, agree or neutral counted as a barrier. For each participant, the identified barriers were summed and calculated as the total number of barriers, as well as number of barriers classified under each of the COM-B categories of capability, opportunity, and motivation.

Data from Excel were then exported to IBM SPSS Version 25 where missing data were checked and excluded. Descriptive statistics were used for the analysis relating to Aims 1–3. Data were reported as means (standard deviations) for continuous variables or frequencies and percentages for categorical variables.

Analysis of variance was used to determine the relationship between demographics and practice of physical activity counselling, with Tukey’s post hoc comparison used to compare between groups (e.g., inpatient, outpatient/community, mixed inpatient/outpatient practice settings) for significant analyses (Aim 4). Linear regression was used to explore the relationship between the total number of elements of physical activity counselling incorporated into practice and the number of barriers reported within each of the three categories (Capability, Opportunity, and Motivation) as well as the total number of barriers (Aim 4). Relative risks (95% CIs) were calculated to explore the relationship between whether or not the physiotherapist identified they had the skill of using different physical activity counselling elements (e.g., objectively assessing physical activity) and whether or not they reported currently using that element in practice (Aim 4).

## 3. Results

### 3.1. Participants’ Characteristics

A total of 87 surveys were submitted across five hospitals in SWSLHD between March and April 2020. Of the 87 surveys, 3 (4%) were initiated but no data entered, so they were excluded, 1 (1%) was partially completed, and 83 (95%) were fully completed, giving a total sample of 84 (52% of the potentially available sample). Table 1 provides detailed demographic characteristics and clinical setting of the sample. Overall, there was good representation across different demographic categories and across 14 areas of physiotherapy practice. Regardless of setting (inpatient, outpatient, mixed), physiotherapists reported an average of at least six occasions of service per patient lasting over 30 min per session (Table 2).

### 3.2. Current Practice of Physical Activity Counselling by Physiotherapists within Routine Care

Figure 2 presents the frequency of physiotherapists using the fifteen different elements of physical activity counselling specified in the survey within their usual healthcare interactions. Physiotherapists reported using on average 5 (SD:3) elements of physical activity counselling with at least 50% of their patients. Seventy percent of the physiotherapists indicated they frequently or often raise or discuss overall physical activity to more than half of their patients. Approximately 60% of the physiotherapists indicated they use language to motivate physical activity behaviour change, and only 20% of physiotherapists provided a self-monitoring strategy to more than half of their patients. The least included components of physical activity counselling were measuring physical activity, contacting physical activity providers, and attending at least one physical activity program with an individual patient.

Table 2 presents the effect of experience, clinical setting, and area of practice on the total amount of physical activity counselling elements in physiotherapy usual care. There was no significant difference between physiotherapists who used more or less elements of physical activity counselling with regards to their years of experience of practice (F (4,78) = 0.8; *p* = 0.5). However, there was a significant difference based on setting (F (2,81) = 7.9; *p* < 0.01), with physiotherapists who primarily saw patients in the inpatient setting incorporating 3.0 (95% CI 1.2 to 4.8, *p* < 0.01) fewer elements of physical activity counselling compared with physiotherapists in outpatient settings. In terms of areas of practice, some physiotherapists worked across multiple areas; therefore, they could not be statistically compared. Nevertheless, physiotherapists working in Emergency Departments, Rehabilitation and Aged Care reported incorporating the most elements, whereas those working in Medical and Surgical wards reported incorporating the least.

### 3.3. Physiotherapist Perceptions of Their Patients’ Readiness for Structured Community Based Physical Activity

Physiotherapists were asked if they believed their patients were ready to be referred to structured community-based physical activity on discharge, or whether they believed they needed further interventions or support from a health professional. Sixty-nine percent of the physiotherapists were unsure or suggested their patients were not ready for direct referral to community-based physical activity, with over 90% of them reporting their patients would benefit from a transitional stage with a health professional-led program (Table 3).

### 3.4. Influences on Physiotherapists’ Ability to Incorporate PA Counselling into Practice within the COM-B Framework, and Its Relationship with Total Number of PA Counselling Elements

Table 4 displays the number (%) of perceived barriers relating to physiotherapists’ capabilities (knowledge, skills), their motivations (beliefs and feelings) and their opportunities (physical and social and including physiotherapist perceptions of patient opportunities) to incorporate physical activity counselling into routine care. Of the 53 potential influences included in the survey, physiotherapists reported an average of 25 (SD:9) barriers that may influence their ability to incorporate physical activity counselling into practice. For every additional reported barrier there were 0.3 (95% CI 0.3 to 0.2) fewer elements of physical activity counselling reported by physiotherapists (Table 4). The greatest proportion of barriers was categorised under the COM-B component of opportunity, followed by capability and then motivation. In terms of opportunity, although most (64%) physiotherapists felt they had time to incorporate physical activity counselling into practice, 57% of physiotherapists indicated insufficient time to locate physical activity opportunities. Most physiotherapists (93–95%) also perceived lack of opportunity for their patients to engage in physical activity due to financial, transport and support barriers. Regarding the knowledge aspect of capability, physiotherapists reported knowledge gaps in the proportion of people meeting physical activity guidelines (75%), the evidence supporting the use of physical activity counselling within healthcare (62%), what patients and physiotherapists think about physical activity counselling within healthcare (71–79%) and the physical activity opportunities available locally (61%). From the skill aspect, at least half of the physiotherapists perceived uncertainty in finding appropriate resources, locating and making a referral to suitable physical activity opportunities, assessing physical activity objectively and setting physical activity action plans. Nevertheless, responses generally indicated good motivation in acknowledging the role of physiotherapists in providing physical activity counselling in routine care (only 1% agreed this was not part of a physiotherapist’s job), although 60% believed that their patients were not willing to discuss physical activity. The full survey results for each influencer are presented in Appendix A.

### 3.5. Relationship between PA Counselling Skills and Incorporating Elements

We further explored the relationship between self-reported skills (an aspect of psychological capability) and the corresponding physical activity counselling elements. For those with good self-reported skills at a particular element (e.g., referring patients to physical activity opportunities), overall, they were more likely to report using the corresponding physical activity counselling element with at least 50% of their patients who could be more active. Where this did not hold true was for assessment of physical activity with physiotherapists reporting having the skills but not using this element in practice. Appendix A presents a full list of relationships between self-reported skills and use of different physical activity counselling elements within usual physiotherapy care.

## 4. Discussion

This cross-sectional study aimed to explore the current practice of physiotherapists incorporating physical activity counselling within routine care and the influences on their ability to do this within five publicly funded hospitals in South Western Sydney, Australia. We found that while a large proportion of physiotherapists raise or discuss physical activity with their patients, only a small number use key strategies to support behaviour change of their patients to become more active. Most physiotherapists perceived their patients not to be ready for direct referral to community-based physical activity and considered that patients would generally require a transition stage or further support from a health professional to engage with community-based physical activity. This study also demonstrated that physiotherapists believe they should incorporate physical activity counselling into routine care, but reported barriers associated with their opportunity, capability, and to a lesser extent their motivation to deliver physical activity counselling. Unsurprisingly, the more barriers the physiotherapist reported, the fewer the elements of physical activity counselling that were used. Physiotherapists who reported skills in using different counselling elements were more likely to use that element in practice. Overall, these findings reflect that physiotherapists believe it is within their scope of practice to deliver physical activity counselling in routine care but are currently not effectively putting this into practice. Key influences identified in this work will guide development of implementation strategies to address this evidence-practice gap in a future planned study.

The elements of physical activity counselling investigated within this survey included behaviour change techniques found in previous studies to positively impact on participants’ physical activity behaviour. In a systematic review conducted by Michie et al. [11] the use of a self-monitoring strategy (e.g., activity diary, activity monitor, pedometer) was shown to be the most effective behaviour change technique at increasing physical activity if combined with at least one of four other self-regulatory techniques (intention formation, goal setting, feedback, and review of behavioural goals). Similarly, a systematic review of physical activity interventions for people with physical disabilities found that interventions that included self-monitoring strategies produced the largest effects [34]. From our survey, only 20% of physiotherapists reported using a self-monitoring strategy and only 55% of them used goal setting with at least half of their patients who could be more active. This limited use of behaviour change techniques by physiotherapists has also been demonstrated in a systematic review which found small numbers of behaviour change techniques used in observational and experimental physiotherapy interventions, with effective interventions more likely to incorporate behaviour change techniques [35]. The findings from this review and our survey strongly indicate that implementation strategies are needed to assist physiotherapists to incorporate behaviour change techniques more effectively within their practice.

An interesting finding from our survey was that many physiotherapists perceived their patients were not ready for direct referral to community-based physical activity and would benefit from a transitional stage within a health professional-led program. This finding aligns with work led by Rimmer et al. in the United States, who proposed models that emphasise the connection between health and community services and incorporate a transition stage from rehabilitation discharge to community physical activity participation [36,37]. Further research is needed to understand why physiotherapists do not perceive patients to be ready for community-based physical activity and what is needed to address this gap. One possible explanation is that physiotherapists may be uncertain about community services and whether exercise leaders have the knowledge, skills, and facilities to cater and potentially tailor physical activity for people who have a physical disability. The newly developed UK Physical Activity Referral Scheme (PARS) Taxonomy [38] provides a structure to detail physical activity service characteristics and referral processes. This structure may be worth testing as a tool for physiotherapists to gather information about local services to enable selection of services that cater for their patients and thus facilitate referrals.

In terms of influences on physiotherapists’ ability to incorporate physical activity counselling into practice, our results were consistent with other published surveys, with physiotherapists acknowledging their role in promoting physical activity [20,22]. Our study further identified capability barriers including a lack of knowledge of evidence supporting physical activity counselling and what local physical activity opportunities exist, and lack of skills in how to measure physical activity, make action plans and locate and make referrals to community services. Several opportunity barriers were also identified, including a lack of physical opportunities in regards to therapist time to locate physical activity opportunities and patient resources. Given the reported lack of knowledge and skills in locating physical activity opportunities, and the feelings of uncertainty of community providers being able to cater for their patients, it is likely a combination of capability, motivation and opportunity limiting referral to community physical activities. Therefore, implementation strategies should include both education to address knowledge gaps and training of physical activity counselling skills (with consideration of patient resources and COVID-19 restrictions), as well as tailored strategies to support local team solutions including building relationships between physiotherapists and local community physical activity providers.

This study has strengths in that it uses a theoretical framework to identify and categorise influencers on a behaviour to enable the development of implementation strategies to address the evidence-practice gap for the local context. With this strength comes the limitation that this study was conducted in a specific context—public hospitals in Australia servicing a multicultural and low socioeconomic population. Thus, the results may not apply to all physiotherapy practice and all populations. However, given the importance of understanding the local context to change behaviour, it is crucial these local behavioural diagnoses are conducted, and lessons can be learnt more broadly regarding the use of theoretical frameworks and processes to change clinician behaviour. Other limitations include the response rate (52% of the population sample frame), which was less than the 60–70% previously recommended [33], due to the impact on our recruitment methods by the COVID-19 pandemic. Our sample includes a good distribution across key demographic and practice variables, indicating the data is representative, and thus can be generalised to the local district. The monetary incentive used to increase response rate might also have introduced a bias to the demographics of attracted participants. However, the incentive was small ($50 gift card) and only mentioned to a minimal extent. Another potential limitation is the effect of recall bias, where physiotherapists may have recalled their frequency of physical activity counselling erroneously, resulting in under-estimating or usually over-estimating the amount of physical activity counselling provided [39]. There may also have been a lack of accuracy of reporting where physiotherapists did not fully understand the specific elements in physical activity counselling (e.g., use of motivational language) and overstating their skill and use of that element. Future research using ethnographic approaches [40] with observations of physiotherapists’ interaction with patients [41] and qualitative interviews may help to capture an in-depth and more accurate picture of physiotherapists’ practice.

## 5. Conclusions

This study demonstrated that physiotherapists employed in public hospitals in a local health district in Australia often raise the topic of physical activity with their patients, but only a small number use key behaviour change strategies to support their patients to become more active, indicating an evidence-practice gap. Influencers on this behaviour included lack of opportunity of therapists to locate appropriate physical activity opportunities and lack of patient opportunities to participate due to transport, finance and support barriers. Other influences were mostly categorised as capability influences and included lack of knowledge about physical activity and counselling, and lack of skills in providing some elements of the counselling. These findings have been used to guide development of implementation strategies to be used in a planned implementation-effectiveness study in this setting to address the evidence-practice gap. The methods and theoretical behaviour change model used within this study can inform future studies aimed at changing clinician behaviour to ensure a thorough behavioural diagnosis is conducted to maximise potential for implementation success.

## Figures and Tables

**Figure 1 ijerph-18-04762-f001:**
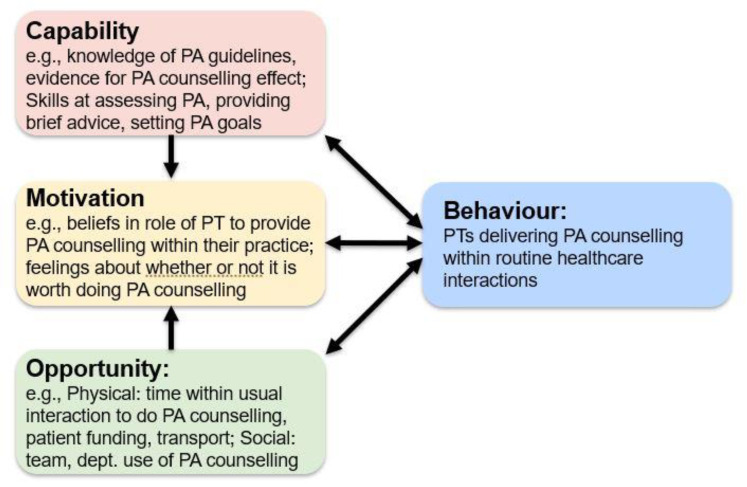
Examples of potential influences on physiotherapists behaviour of providing physical activity counselling framed within the COM-B theoretical model of behaviour change. Key: PA: physical activity; PT: physiotherapist; dept.: department.

**Figure 2 ijerph-18-04762-f002:**
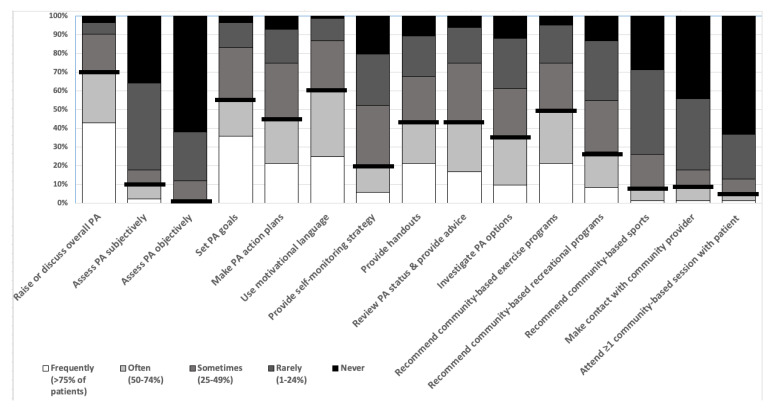
Current PA counselling in routine physiotherapy care. Key: Bold horizontal lines indicate the percentage of physiotherapists who reported providing each physical activity counselling element to more than 50% of patients who the physiotherapists considered could be more active; PA: physical activity.

**Table 1 ijerph-18-04762-t001:** Characteristics of participants.

Characteristic		*n* = 84
Age (yr), *n* (%)		
<25		19 (23)
25–34		40 (48)
35–44		18 (22)
45 and over		6 (7)
Gender, number female, *n* (%)		66 (79)
Years practised as a physiotherapist (years)		
0–2		17 (20)
2–5		23 (28)
5–8		13 (16)
8–12		10 (12)
>12		20 (24)
Hospital, *n* (%)		
Hospital 1		44 (52)
Hospital 2		18 (21)
Hospital 3		14 (17)
Hospital 4		7 (8)
Hospital 5		1 (1)
Employment status, *n* (%)		
Permanent staff		55 (67)
Contract staff		17 (21)
First year graduate program		10 (12)
Full time equivalent, *n* (%)		
Full-time		65 (78)
Part-time		18 (22)
Classification of position, *n* (%)		
Level 1–2: Junior clinician		47 (57)
Level 3–4: Senior clinician		26 (32)
Level 5–6: Health Professional Educator/Clinical Specialist		5 (6)
Manager		4 (5)
Position, *n* (%)		
Rotational position		43 (53)
Non-rotating position		38 (47)
Practice setting, *n* (%)		
Inpatient		43 (51)
Outpatient/Community		29 (35)
Mixed inpatient and outpatient		12 (14)
Area of physiotherapy, *n* (%) *		
Musculoskeletal/Rheumatology/Hands		23 (27)
Rehabilitation		21 (25)
Orthopaedics		17 (20)
Aged Care		17 (20)
Cardiopulmonary		14 (17)
Intensive Care		11 (13)
Emergency Department		10 (12)
Surgical		10 (12)
Medical		10 (12)
Cancer		6 (7)
Women & Men’s health		6 (7)
Acute Neurological		6 (7)
Other (e.g., paediatrics, renal, chronic disease, palliative care)		8 (10)
Current workload		
Mean (SD) number of patients seen in a typical workday	Inpatient	10.2 (2.6)
	Outpatient	8.0 (4.5)
	Mixed	6.5 (2.3)
Mean (SD) time spent with patient per session (min)	Inpatient	32.8 (11.6)
	Outpatient	46.7 (20.3)
	Mixed	46.8 (11.7)
Mean (SD) number of occasions of service before discharged	Inpatient	8.8 (12.8)
	Outpatient	6.4 (3.9)
	Mixed	8.4 (14.1)
Mean (SD) number of new patients seen per week	Inpatient	10.8 (7.2)
	Outpatient	9.2 (11.3)
	Mixed	9.6 (8.7)
Training		
Attended training on behaviour change/ motivational interviewing/ health coaching, yes, *n* (%)		30 (36)

* Physiotherapists could select more than one current area of physiotherapy practice.

**Table 2 ijerph-18-04762-t002:** The influence of experience and clinical setting on the total number of elements in physiotherapy routine care.

Demographic Categories	Mean (SD) Number of PA Counselling Elements Used
PT years of experience	
<2 years (*n* = 17)	4.0 (3.0)
2 ≤ 5yrs (*n* = 23)	5.0 (3.6)
5 ≤ 8yrs (*n* = 13)	5.7 (3.5)
8 ≤ 12yrs (*n* = 10)	3.8 (3.4)
>12 years (*n* = 20)	5.1 (3.6)
Setting	
Inpatient (*n* = 43)	3.6 (3.0)
Outpatient/community (*n* = 29)	6.6 (3.5)
Mixed inpatient/outpatient (*n* = 12)	5.3 (2.7)
Area of physio practice *	
MSK; Rheum; Hands (*n* = 19)	5.1 (2.9)
Orthopaedics (*n* = 17)	4.8 (3.2)
Rehabilitation (*n* = 21)	5.7 (3.8)
Aged care (*n* = 17)	5.5 (4.1)
Cardiopulmonary (*n* = 14)	4.9 (3.2)
ED (*n* = 10)	6.7 (3.5)
Cancer (*n* = 6)	4.2 (3.4)
Medical (*n* = 10)	2.7 (1.7)
Women and Men’s Health (*n* = 6)	5.3 (3.6)
Surgical (*n* = 10)	3.7 (2.9)
ICU (*n* = 11)	4.5 (3.2)
Neuro (*n* = 6)	4 (3.3)
Other: Paediatrics, Renal, Palliative (*n* = 8)	3.5 (2.4)

Key: * Physiotherapists worked across areas of practice and therefore may be represented in more than one area; PA: physical activity; PT: physiotherapy; MSK: Musculoskeletal, Rheum: Rheumatology, ED: Emergency Department, ICU: Intensive Care unit, Neuro: Neurological.

**Table 3 ijerph-18-04762-t003:** Physiotherapist perceptions of patients’ readiness for structured community-based physical activity (*n* = 84).

	Yes *n* (%)	No *n* (%)	Unsure *n* (%)
Ready to be referred directly to community PA programs	26 (32)	44 (54)	12 (15)
Would benefit from a transitional stage with a health professional-led program	76 (90)	2 (2)	6 (7)
Require further treatment from a health professional prior to referral	59 (71)	12 (14)	12 (14)
Benefit from supported introduction or extra advice about community PA programs from a health professional	79 (94)	2 (2)	3 (4)

Key: PA: physical activity.

**Table 4 ijerph-18-04762-t004:** Number of barriers to physical activity counselling reported by physiotherapist by aspects of the COM-B framework and their relationship with the total number of elements of PA counselling used (*n* = 84).

	Mean (SD)	Percentage of Total Barriers Reported in Each Category
Total barriers reported (0–53)	25.1 (8.7)	47
Capability (0–28)	13.3 (5.7)	47
Knowledge (0–14)	7.6 (2.6)	54
Cognition (0–13)	5.0 (3.8)	39
Opportunity (0–12)	7.1 (2.9)	59
Physical (0–7)	4.7 (1.8)	67
Social (0–5)	2.4 (1.7)	47
Motivation (0–13)	4.8 (2.9)	37
	Number (95% CI) PA counselling elements *	*p* value
Total barriers	−0.3 (−0.3 to −0.2)	<0.01
Barriers in Capability	−0.3 (−0.4 to −0.2)	<0.01
Barriers in Opportunity	−0.2 (−0.4 to 0.1)	0.21
Barriers in Motivation	−0.2 (−0.4 to 0.1)	0.13

Key: * linear regression analysis to determine the relationship between number of barriers reported and the number of counselling elements reported by physiotherapists used in routine care. A negative co-efficient indicates less counselling element used when more barriers were reported.

## Data Availability

The data presented in this study are available on request from the corresponding author. The data are not publicly available due to ethics approval requirements.

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
