# Peer review of "Current Practice of Physical Activity Counselling within Physiotherapy Usual Care and Influences on Its Use: A Cross-Sectional Survey"

_ijerph, 2021, doi:10.3390/ijerph18094762_

Round 1

Reviewer 1 Report

My comments have been addressed. Thank you very much for your revisions. 

Author Response

Thank you very much for your review and giving us opportunity to improve the manuscript. 

Reviewer 2 Report

After to read carefully the manuscript version R1 this paper continue shows seriuos problems related to scientific procedure in spite of the modifications

Authors has not response all reviewers requeriments detailed in the review inform

This manuscript cannot be accepted as its current form and format. There are some problematic areas of the manuscript and the authors were not able to deal with the essential aspects of so-called “scientific research.”

 Isn’t the test-rest reliability enough in order to measure this parameters after to intervention process?

How authors were able to affirm  that the reliability of the measurements  are related  to consueling and phisiotherapy? How the procedures can be replicated by future studies?

The survey results might not be considered valid as scientific research,at least with regard to this context

Round 2

Reviewer 2 Report

After to read carefully the present manuscript. I recomend to publish it in present way. Due to the fact that authors have  adresed all requeriment in the correct way

This manuscript is a resubmission of an earlier submission. The following is a list of the peer review reports and author responses from that submission.

Round 1

Reviewer 1 Report

This is a cross-sectional study that is investigating the incorporation and effects of physical activity counseling on patients receiving physiotherapy.

Title: Wording is a little confusing. It can be reworded to be more clear on the objective.

Introduction: There needs to be more explanation on the barriers of implementation and how specific interventions can work to overcome these barriers. The COM-B model is explained however, the explanation has too much jargon, it is hard to understand and connect to the research questions.

Methods: Since physical activity levels are significantly lower in this area, perhaps the results would not be generalizable to another population. How many physiotherapists were asked to participate? The authors need to state how the survey was made, was this a group process? How did the authors ensure that the person contacted was the one completing the questionnaire online (additional confidentiality measures taken online) ? If a participant had a question about the survey, how was this managed? 

Results: Sample frame was from 100 to 116 but only 87 surveys were submitted. How many surveys were sent? How many people agreed to be part of the study? Table 2 is showing us an association between years of experience as a physiotherapist and the total number of elements in physio-therapy routine care but it is not showing us a visual of this association. Too many tables in the results section make it a little hard to get a full picture of the findings.

Discussion: Good discussion section that summarizes the findings and relates them to barriers of implementation while mentioning limitations of the study. One additional limitation could be the monetary incentive that was offered to physiotherapists, creating a biased sample. One interesting thing that could be added is perhaps the impact of covid-19 on the implementation of such interventions since covid-19 was mentioned earlier as an obstacle the research team had to overcome.

Conclusion: The conclusion is not very clear. What elements specifically influence the use of physical activity counseling?

Writing style: Some sentences are very long and use many long words that make them hard to understand. Also many tables in the results section without clear themes being communicated to the reader. Editorial assistance would be beneficial before publication.

Reviewer 2 Report

Dear Authors,

Glad to have an opportunity to review this manuscript, but first of all, this manuscript cannot be accepted as its current form and format. There are some problematic areas of the manuscript and the authors were not able to deal with the essential aspects of so-called “scientific research.”

Could be interesting to readers of International Journal of Environmental Research and Public Health. However, from my point of view, authors should include the following requeriments

There are several methodological concerns that limit the reader's understanding. Below I have provided comments for the authors.

The authors have investigated about Physical activity counselling within physiotherapy usual care and influences on its use. There are several methodological concerns that limit the reader's understanding of why this experiment was conducted. Below I have provided comments for the authors.

What is a point to Physical activity counselling?

While a clinimetric tool employed can be considered as  a valid measure for health aspects, what is the sense of use this survey to  ? Isn’t the test-rest reliability enough in order to measure this parameters after to intervention process? What is the advantage to use this survey? What was the reason for considerer useful only in physiotherapy I bellive that  authors should compere authors compare in different rehabilitation clinician  groups?

Please justify this topic.

Moreover, a more exhaustive review of state of art is needed, current study can be improved

Introduction section may be improved adding new information in order to provide an adequate state-of-the-art including some references.

 Methods are not well-designed with relevant and complete information. How authors did determine the sample size calculations?, In adition threre’s no good description of the properties of the outcome measurements as well as  were not detailed statistical analyses included.

For example, tables, and redaction of the results present several mistakes due to the fact significance are not included, In summary, results does not provide a good presentation of the main finding of the study.

Moreve authors  must include a reference to Ethics requirements Helsinki declaration and Strobe methods, and should include the following reference properly to justify this topic

-Vandenbroucke, J.P.; von Elm, E.; Altman, D.G.; Gøtzsche, P.C.; Mulrow, C.D.; Pocock, S.J.; Poole, C.; Schlesselman, J.J.; Egger, M.; STROBE Initiative Strengthening the Reporting of Observational Studies in Epidemiology (STROBE): explanation and elaboration. Int. J. Surg. 201412, 1500–24.

-Holt, G.R. Declaration of Helsinki—The World’s Document of Conscience and Responsibility. South. Med. J. 2014107, 407–407.

Discussion section may include future research studies secondary to the current findings of this study. Clinical considerations, limitations and overall discussion are well-presented, but future research may be useful in order to propose future research regarding this field.

How authors were able to affirm  that the reliability of the measurements  are related  to consueling and phisiotherapy? How the procedures can be replicated by future studies?.

What is the reason beacause authors use the  samples?, For this reason can be considered a risk of bias due to the fact sample size are not properly calculate

Using descriptive statistics test is not de adequate statistical analyse to consueling and phisyotherapy interevention, specially if sample size is short, when statistical differences appears, for this reason pheraps authors should revise statistical analyse for example using another test in order to improve their achiviment

Round 2

Reviewer 1 Report

My comments have been addressed. Thank you very much for the revisions.

Author Response

Thank you for your suggestions and commentaries.

Reviewer 2 Report

I want to thank you for the opportunity to review this manuscript. After the review and in my humble opinion, from my point of view, the manuscript presents major problems that inaceptable for publication:

In their first revision of manuscript, the authors have not addressed my questions/comments properly.

Even though I had indicated exactly what should they do, authors have not completed my requirement for this reason I suggest they need to improve their manuscript in order to increase the manuscript quality

For example, I do not why they have not included the references with regard to ethics procedure or why do not estimated my suggest regarding statistical analyse. In fact, if they do not have intention to change their method, I think this research will not reach the minimum standard to be published, due to the fact that consueling have not enough scientific evidence level, from my point of view

In spite of all the improvements that I transfer to authors in my review comments are necessary, if you disagree, please withdraw my review comments. In their first revision of manuscript, the authors have addressed my questions/comments properly
